# The Role of Plasminogen Activator Inhibitor 1 in Predicting Sepsis-Associated Liver Dysfunction: An Observational Study

**DOI:** 10.3390/ijerph20064846

**Published:** 2023-03-09

**Authors:** Ewa Woźnica-Niesobska, Patrycja Leśnik, Jarosław Janc, Małgorzata Zalewska, Lidia Łysenko

**Affiliations:** 1Department of Anaesthesiology and Intensive Therapy, Wroclaw Medical University, 50-556 Wroclaw, Poland; 2Department of Anaesthesiology and Intensive Therapy, 4th Military Clinical Hospital, 50-981 Wroclaw, Poland; 3Department of Infectious Diseases Liver Diseases and Acquired Immune Deficiencies, Wroclaw Medical University, 50-367 Wroclaw, Poland

**Keywords:** sepsis-associated liver dysfunction (SALD), liver failure, plasminogen activator inhibitor 1 (PAI-1), sepsis

## Abstract

Sepsis-associated liver dysfunction (SALD) is associated with a poor prognosis and increased mortality in the intensive care unit. Bilirubin is one of the components of Sequential Organ Failure Assessment used in Sepsis-3 criteria. Hyperbilirubinemia is a late and non-specific symptom of liver dysfunction. This study aimed to identify plasma biomarkers that could be used for an early diagnosis of SALD. This prospective, observational study was conducted on a group of 79 patients with sepsis and septic shock treated in the ICU. Plasma biomarkers—prothrombin time, INR, antithrombin III, bilirubin, aspartate transaminase (AST), alanine transaminase, alkaline phosphatase, gamma glutamyl transferase, albumin, endothelin-1, hepcidin, plasminogen activator inhibitor-1 (PAI-1), thrombin-antithrombin complex, and interferon-gamma inducible protein (10 kDa) were analysed. Plasma samples were obtained within 24 h after having developed sepsis/septic shock. Enrolled patients were followed for 14 days for developing SALD and 28 days for overall survival. A total of 24 patients (30.4%) developed SALD. PAI-1 with a cut-off value of 48.7 ng/mL was shown to be a predictor of SALD (AUC = 0.671, sensitivity 87.3%, and specificity 50.0%) and of 28-day survival in patients with sepsis/septic shock (*p* = 0.001). Measuring PAI-1 serum levels at the onset of sepsis and septic shock may be useful in predicting the development of SALD. This should be verified in multicenter prospective clinical trials.

## 1. Introduction

Sepsis-associated liver dysfunction (SALD) is associated with a poor prognosis and increased mortality in Intensive Care Unit (ICU) patients [1]. It remains a key component of multiple organ dysfunction syndrome (MODS), which complicates sepsis. In sepsis, infection, combined with the hyperactivity of the inflammatory response and microcirculatory failure, contribute to organ dysfunction. In available reports, due to the lack of homogenous diagnostic criteria, the incidence of SALD varies from 1.1% up to 34.7% [2]. It may present as hypoxic hepatitis (HH), sepsis-induced cholestasis, and/or coagulopathy [3], and the degree of organ injury may range from mild liver dysfunction to life-threatening liver failure. According to Sepsis-3 criteria, an acute change in a sequential organ failure assessment (SOFA) score of ≥2 in response to an infection allows life-threatening organ dysfunction, defined as sepsis, to be diagnosed [4]. An increase in bilirubin serum concentration can be a signal of developing liver dysfunction. The elevation in bilirubin concentration in human plasma of >1.9 mg/dL gives a SOFA score of 2. Bilirubin itself is a late and non-specific marker of hepatic dysfunction. Hyperbilirubinemia may be caused by hemolysis, cholestasis, and hepatic dysfunction of multiple origins: decreased bilirubin transport, uptake, and clearance, as well as hepatic ischemia and hepatocellular damage [2].

Acute liver failure (ALF) is defined as an acute liver injury with the onset of hepatic encephalopathy (HE) and an increase in the International Normalised Ratio (INR) of >1.5 in patients with no pre-existing liver disease [5]. In an ICU setting, due to the multiple possible causes of impaired consciousness, diagnosing HE is difficult. Sepsis-induced HH is a rare cause of ALF, occurring mainly due to organ hypoperfusion. However, increased blood flow and cardiac output in the primary stages of septic shock might not be enough to compensate for an increased hepatic oxygen demand and result in the development of HH [2]. Ischemic/hypoxic hepatitis is reported in up to 2.5% of all ICU patients [6].

We postulated that the definition of SALD requires redefining and creating more specific diagnostic criteria. The aim of the study was to identify plasma biomarkers that could be used for an early diagnosis of SALD. The secondary goal was to assess the impact of selected variables on mortality, including the occurrence of SALD.

## 2. Materials and Methods

### 2.1. Design and Settings

A single-center prospective observational study was conducted in a 14-bed medico-surgical ICU at Wroclaw University Hospital from September 2015 to April 2019. Taking SOFA scoring into consideration, we defined SALD as an acute elevation of the serum bilirubin level of 2 mg/dL or more, excluding causes of hyperbilirubinemia other than sepsis.

### 2.2. Ethics

The study protocol was approved by the Bioethics Committee of the Wroclaw Medical University (approval numbers KB 415/2015 and KB 670/2017). Informed written consent was obtained from all the patients or their families. The study was carried out in accordance with the guidelines of the Declaration of Helsinki and good clinical practice.

### 2.3. Participants

The study enrolled adult patients admitted to the ICU due to sepsis/septic shock and patients who developed sepsis/septic shock during their ICU stay. Sepsis/septic shock was defined according to the Sepsis-3 criteria [4]. Exclusion criteria were age < 18 years old, pregnancy, pre-existing liver disease (cholestatic disorders; genetic, vascular, and metabolic liver diseases; viral hepatitis; liver tumors); liver cirrhosis (Child-Pugh class A, B, or C); immunosuppression; HIV infection; cancer; and xenobiotic intoxication. Patients with missing data were excluded from the study. All patients were treated according to the Surviving Sepsis Guidelines 2012 (patients admitted to the ICU before the publication of the new definition and guidelines) and the Surviving Sepsis Guidelines 2016 [4,7].

Once the patient was qualified for the study, clinical and demographic data, including age, sex, comorbidities, the Acute Physiology and Chronic Health Evaluation (APACHE II) score, the SOFA score, and the origin of sepsis, were recorded in the study protocol. The SOFA score was subsequently recorded on days 1, 3, 5, 7, and 14. Patients were also screened for the development of Disseminated Intravascular Coagulation (DIC) using ISTH Criteria for DIC on the same days [8].

The primary endpoint of the study was the development of SALD while staying in the ICU. The secondary endpoint was 28 day overall survival.

### 2.4. Outcomes

Plasma biomarkers (prothrombin time, INR, antithrombin III, bilirubin, aspartate transaminase-AST, alanine transaminase-ALT, albumin, alkaline phosphatase-ALP, gamma glutamyl transpherase-GGT, lactate levels) were measured within 24 h after enrolment in the study (day 1) and on consecutive days 3, 5, 7, and 14. An abdominal ultrasound was performed to exclude mechanical causes of hyperbilirubinemia. At the same time, blood samples were drawn for the analysis of novel biomarkers: endothelin-1 (ET-1), hepcidin, plasminogen activator inhibitor 1 (PAI-1), the thrombin-antithrombin (TAT) complex, and interferon-gamma inducible protein 10 kDa (IP-10).

Blood samples were collected and centrifuged at 3000 rpm at room temperature in EDTA tubes; consecutively, samples were frozen at −28 °C within 30 min of collection. Levels of human hepcidin IP-10, PAI-1, and ET-1 were measured using Quantikine ELISA tests (R&D Systems Inc., Minneapolis, MN, USA). Human TAT complex levels were measured using Assay Max (Assaypro LLC, St. Charles, MO, USA). All assays employ the quantitative sandwich enzyme immunoassay technique. Tests were performed and interpreted according to the manufacturer’s instructions.

### 2.5. Analysed Biomarkers

#### 2.5.1. IP-10

IP-10 is induced in different cells, i.e., leukocytes (neutrophils, monocytes, and macrophages), in response to type 1 and 2 interferons (IFN) and lipopolysaccharide (LPS) stimulation. IP-10 induces apoptosis, cell growth inhibition, chemotaxis, and angiostasis [9]. IP-10 activates the C-X-C motif chemokine receptor 3 (CXCR3) in response to viral infections, autoimmune diseases, allotransplantation, and cancer, which is an important regulator of natural killer (NK), natural killer T (NKT), and T helper (Th)1 lymphocyte trafficking [10]. In the liver, IP-10 is secreted by hepatocytes in areas of lobular inflammation and may be responsible for the development of intrahepatic inflammation. IP-10 is involved in the pathogenesis of hepatitis C and hepatitis B and has recently been shown to play a pivotal role in the pathogenesis of experimental steatohepatitis [11,12,13].

#### 2.5.2. ET-1

Elevated ET-1 plasma levels have been reported in patients with septic shock [14]. ET-1 is a vasoconstrictive molecule synthesised by endothelial cells in response to its injury. It also stimulates phagocytosis and chemotaxis of monocytes/macrophages, and neutrophils, therefore augmenting the inflammatory response [15]. In a murine model, hepatic macrophages were shown to be the primary source of elevated plasma levels of ET-1 [16]. Hepatic stellate cells, sinusoidal endothelial cells, and Kupffer cells are the hepatic sources of ET-1. ET-1, being a local regulator of hepatic sinusoidal microcirculation, acts via ETA/ETB2 receptors on hepatic stellate cells, causing liver sinusoid constriction [17]. Endothelin-mediated microcirculatory failure leads to hepatocellular injury via worsening oxygen delivery and metabolic dysfunction during sepsis [18].

#### 2.5.3. Hepcidin

Hepcidin is a peptide of mostly hepatic origin involved in iron metabolism. Increased serum hepcidin levels were observed in neoplastic diseases, inflammation, and sepsis [19]. Its expression is suppressed by iron deficiency, anemia, and hypoxia, but induced by iron overload, inflammatory stimuli, and LPS [20]. During inflammatory states, hepcidin expression is induced via the cytokine IL-6 [21]. The antimicrobial function of hepcidin was shown in several studies but the mechanism of both antibacterial and antifungal action is not yet well understood [19]. Iron metabolism may be a useful predictor of outcome in patients with liver disease. In the murine N-acetyl-p-aminophenol (APAP)-induced acute liver failure model, hepcidin was shown to be an independent predictor of mortality [22].

#### 2.5.4. TAT

TAT, a marker of thrombin generation, forms following the neutralisation of thrombin by antithrombin III (AT III) [23]. AT III is synthesised in the liver and is a natural anticoagulant. Its anti-inflammatory function is due to the neutralisation of thrombin, which is responsible for leukocyte rolling and adhesion, but it also depends on blocking the effect of protease-activated receptor-1 [24]. TAT can be used as a sensitive parameter for the latent activation of the clotting pathway. The rise of TAT suggests continuous thrombin generation and antithrombin depletion [25]. In patients with liver cirrhosis, elevated TAT levels are also observed in patients with Child-Pugh A. TAT and AT III are thought to be independently associated with the occurrence of liver dysfunction [26].

#### 2.5.5. PAI-1

PAI-1 is a principal inhibitor of fibrinolysis. Elevated PAI-1 levels have been linked to sepsis-induced coagulopathy and the development of disseminated intravascular coagulation (DIC). PAI-1 levels correlate with the severity of MODS in sepsis and DIC [27]. The PAI-1 gene is expressed in the liver, endothelial cells, macrophages, adipose tissue, the heart, and kidney [28]. An increase in plasma PAI-1 after LPS stimulation may be a combined effect of both PAI-1 release from activated platelets and its synthesis associated with PAI-1 gene expression on hepatocytes [29]. Plasma PAI-1 levels are also strongly related to liver steatosis, which supports the hypothesis that the liver is an important source of circulating PAI-1 [30].

### 2.6. Statistical Analysis

Sample size analysis. The sample size was estimated based on our preliminary study results, evaluating the difference in PAI-1 levels between the SALD and no-SALD groups, and was calculated based on a *t*-test of two-sample means (independent-sample *t*-test). The alpha level was set at 0.05 and the test power at 0.8. It was also assumed that the assessed variable was not correlated, and a two-tailed null hypothesis was adopted. In the analysis of sample size estimation (PAI-1 level), the means and standard deviations of the PAI-1 level from the SALD and no-SALD groups were used. Based on the results, an estimated sample size of at least n = 24 participants in each group was obtained. Continuous data are presented as the median and lower and upper quartiles for non-normally distributed variables or as the mean and standard deviation for normally distributed variables. The statistical differences between the groups were calculated using the non-parametric Mann–Whitney U test. Statistical significance between the frequencies was calculated using the chi-square test. The relation between the two parameters was assessed using a correlation analysis, and the Spearman correlation coefficients were calculated. Survival curves were obtained using the Kaplan–Meier method and were compared using the log-rank test. The multivariate analysis was performed using the Cox proportional hazard regression model. The receiver operating characteristic (ROC) curve analysis was performed to calculate the area under the receiver operating characteristic curve. The best cut-off values were calculated to maximise the Youden index. The positive predictive value (PPV), negative predictive value (NPV), and accuracy (true positive + true negative/N) were also calculated. A P value of less than 0.05 was required to reject the null hypothesis. A statistical analysis was performed using the Statistica 13 software (TIBCO Software Inc., Palo Alto, CA, USA).

## 3. Results

A total of 79 patients were included in the study. The flow of patients is shown in Figure 1.

Patients were severely ill, with a mean APACHE II score of 24. The DIC score according to the ISTH criteria was calculated. None of the patients in the study group developed DIC at any time during observation.

Table 1 shows the characteristics of the groups at the time of inclusion and 28 day survival, together with a comparison of these characteristics between the SALD group and the no-SALD group.

### 3.1. Comorbidities and Source of Sepsis

#### 3.1.1. Comorbidities

There was no statistically significant difference in comorbidities between the groups. None of the patients in our study group had metabolic syndrome. In our study group, neither alcohol abuse nor obesity had a statistically significant impact on the development of SALD. Table 2 shows the comorbidities of the patients and their comparison between the SALD group and the no-SALD group.

#### 3.1.2. Source of Sepsis

Apart from bloodstream infections (BSI), there was no difference in the source of sepsis between the groups. There was no case of BSI in the no-SALD group. Table 3 shows the source of the sepsis and its comparison between the SALD group and the no-SALD group.

### 3.2. Routinely Measured Biomarkers

The plasma biomarker levels at the time of enrollment in the study (day 1) are shown in Table 4. AST, lactate level, bilirubin, and ALP were significantly higher in the SALD group compared to the no-SALD group. What is more, statistically significantly higher levels of procalcitonin were observed in the SALD group compared to the no-SALD group (28.0 ng/mL (3.1 ÷ 71.3) vs. 4.0 ng/mL (1.7 ÷ 17.8), *p* = 0.014). 

### 3.3. Analysed Biomarkers

At baseline, we observed a statistically significant difference between the groups only for PAI-1 levels. In the no-SALD group, the levels of PAI-1 were significantly lower than in the SALD group. The levels of analysed biomarkers at the time of enrolment in the study (day 1) are shown in Table 5.

Because there was no statistically significant difference between the SALD and no-SALD groups for other biomarkers, only PAI-1 was taken for further analysis.

### 3.4. PAI-1

#### 3.4.1. PAI-1 Levels

In the no-SALD group, the levels of PAI-1 were significantly lower than in the SALD group. The statistically significant difference between the groups remained until day 7 of observation (Figure 2).

#### 3.4.2. PAI-1 as a Predictor of SALD

The PAI-1 cut-off was calculated using the Youden method. PAI-1 values of ≥48.73 ng/mL (Youden index = 0.41) can be a predictor of SALD, with a sensitivity of 87.3%, specificity of 50.0%, PPV of 80.0%, and NPV of 63.2% (accuracy of 75.9%). Figure 3 shows a ROC curve for PAI-1.

#### 3.4.3. PAI-1 as a Predictor of Mortality

We postulated that bilirubin levels at baseline are not a good predictor of 28 day survival in patients with sepsis and septic shock (Figure 4a). PAI-1 seems to be a good predictor of 28 day survival in patients with sepsis/septic shock (Figure 4b).

Univariate and multivariate Cox proportional hazard regression models were used to investigate factors associated with mortality. Variables with *p* < 0.30 in the univariate analysis were entered into the multivariate model. Variables that were found to be significant (*p* < 0.05) in both the univariate and multivariate analyses were considered to be factors associated with mortality (Table 6).

A multivariate analysis has shown that among analysed biomarkers (Age, PAI-1, APACHE II, SOFA, and Lactate (mmol/L), CRP (mg/L), PCT (ng/mL), BIL (mg/dL), AST (U/L), ALP (U/L), PAI-1 (quantitative variable or categorised variables ref. ≤ 48.73 ng/mL), Sex (ref. M), SALD (ref. no.), only age (HR = 1.04, *p* = 0.002), and lactate (HR = 1.18, *p* < 0.001) can be independent predictors of mortality in the whole group.

## 4. Discussion

Sepsis-associated liver dysfunction (SALD) is associated with a poor prognosis and increased mortality in Intensive Care Unit (ICU) patients [1]. The lack of a generally accepted definition and diagnostic criteria for SALD makes it difficult to estimate its incidence and to compare the results between the studies.

The results of our study show that almost one third (30.4%) of septic patients had an acute elevation of bilirubin up to 2 mg/dL or more. The results are consistent with available reports [2]. Two thirds of the enrolled patients fulfilled our SALD criteria within 24 h of being diagnosed with sepsis and septic shock. 

After analysing our study group demographics, we demonstrate that, contrary to Kobashi H et al. [1], neither sex nor age contributed to SALD development. The difference may be due to different SALD definitions chosen by the authors. Metabolic syndrome, obesity, and alcohol intake are known risk factors increasing the probability of acute liver damage [31]. None of the patients in our study group had metabolic syndrome. Alcohol abuse or obesity did not influence the development of SALD. Other comorbidities did not influence the development of SALD. No difference was observed between the development of SALD and the occurrence of sepsis or septic shock.

Significantly higher levels of PCT were observed in the SALD group compared to the no-SALD group (4.0 vs. 28.0 ng/mL), even though the cause of sepsis (apart from BSI) did not influence the development of SALD. This requires further investigation.

Interestingly, there was no difference in the severity of the disease (APACHE II score) or the degree of organ dysfunction (SOFA score) at the time of inclusion between patients who developed SALD and those who did not. This suggests that hyperbilirubinemia may just be a signal of worsening liver function, which contributes to adverse outcomes via independent pathways (not included in APACHE II scoring) or that the serum bilirubin cut-off value of 2 mg/dL is too low to identify patients with an adverse outcome. This is reflected in the results of our study.

Mild elevations in bilirubin levels, and consequently the degree of organ dysfunction, were not severe enough to cause a difference in mortality between the groups. What is more, there is a wide range of other causes influencing mortality in sepsis/septic shock, other than hyperbilirubinemia itself. This is also confirmed by the fact that there was no difference in APACHE II scoring between our study groups.

Out of the five biomarkers analysed, only PAI-1 at the time of enrollment could be useful to predict the development of SALD. PAI-1 levels peaked at the onset of sepsis and decreased over the course of observation. This is a follow-up to the results of Koyama et al. [32] and Katayama et al. [33]. 

Our study was the first to evaluate the correlation between PAI-1 and liver function in septic patients. In various studies, the liver has been proven to be an important source of circulating PAI-1 [28,30].

Bilirubin as a single biomarker is a poor factor in distinguishing newly developing organ dysfunction from a pre-existing one. Due to its late increase and low specificity, finding a single cause of hyperbilirubinemia in ICU patients remains a challenge. PAI-1 can be a better marker than bilirubin for predicting organ dysfunction, as higher PAI-1 levels were related to a higher SOFA score and lactate level at the onset of sepsis in our study (Appendix A). This is a follow-up to the results of Hoshino et al. [34,35].

The study revealed that the cut-off value for PAI-1 of 48.73 ng/mL was associated with an acute increase in bilirubin levels of 2 mg/dL or more. Our results have shown the low specificity of PAI-1 with a high positive predictive value. The low specificity of PAI-1 may be a result of many origins and different factors inducing its synthesis. An increase in plasma PAI-1 after LPS stimulation may be a combined effect of both the PAI-1 release from activated platelets and its synthesis associated with PAI-1 gene expression on hepatocytes [29]. The results obtained in our study group may indicate that the activated platelets could play a greater role in the increase in PAI-1 levels than its hepatic source does. There must be other risk factors influencing the development of SALD, which require further investigation.

Higher lactate levels, as well as more advanced age, were significant risk factors for mortality, while higher PAI-1 levels could have also been considered as an independent predictor of increasing the risk of death.

Other authors have also investigated PAI-1 as a biomarker of sepsis, organ dysfunction [33,36,37], and DIC [36,37].

Hoshino et al. have shown in their study that patients with sepsis and PAI-1 levels of ≥83 ng/mL have elevated risks of coagulopathy, organ failure, and mortality [35]. Koyama et al. [32] investigated PAI-1 combined with TAT and protein C as predictors for the development of overt DIC. In our cohort, none of the patients developed DIC, so it was impossible to compare these results with ours. The incidence of DIC in septic patients diagnosed with ISTH criteria is reported to be approximately 29% [37]. The authors think that the cause of the lack of overt DIC in the study group was a result of the inclusion criteria used, which resulted in choosing only 79 out of 480 (28.9%) septic patients treated in our ICU. In terms of survival, Koyama’s [32,33], Hoshino’s [35], and Madoiwa’s [36] results were compatible with ours, confirming PAI-1 as a predictor of survival in septic patients, although the PAI-1 cut-off value differed between our study (48.73 ng/mL) and the studies by Madoiwa [36] (90 ng/mL), and Hoshino [35] (83 ng/mL).

PAI-1 may be a marker for prognosing SALD, but due to its low specificity, it must be analysed in combination with other markers. Further studies are needed to find a unique biomarker of SALD, but first it is important to broaden knowledge on its pathophysiology, find an unanimous definition, and define clear diagnostic criteria.

The limitations of the study were that it was a single-center study, and the number of patients enrolled in the study was relatively small, although our results (SALD incidence) matched those demonstrated in previous studies. The lack of a generally accepted SALD definition remained a challenge.

## 5. Conclusions

Measuring PAI-1 serum levels at the onset of sepsis and septic shock may be useful in predicting the development of SALD. In addition, higher lactate levels, as well as more advanced age, are significant risk factors for mortality, while higher PAI-1 levels can also be considered an independent predictor of increasing the risk of death. This should be verified in multicenter prospective clinical trials.

## Figures and Tables

**Figure 1 ijerph-20-04846-f001:**
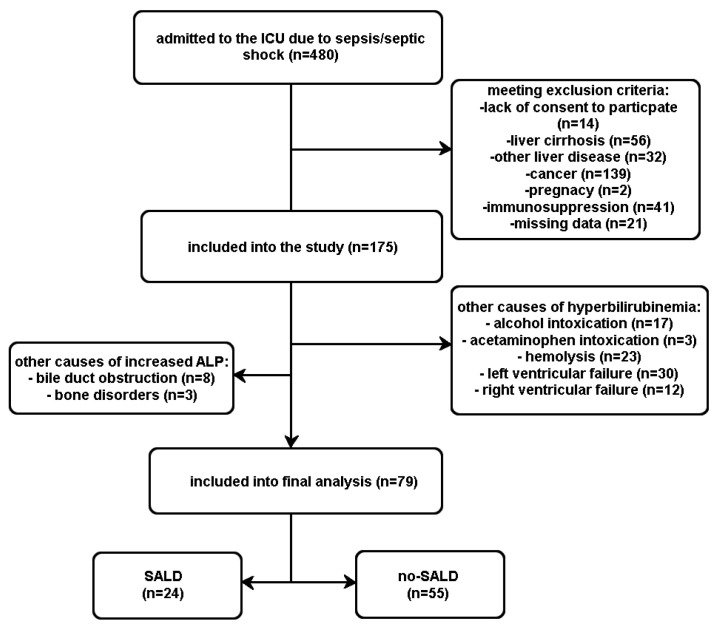
Flow of the patients included in the study.

**Figure 2 ijerph-20-04846-f002:**
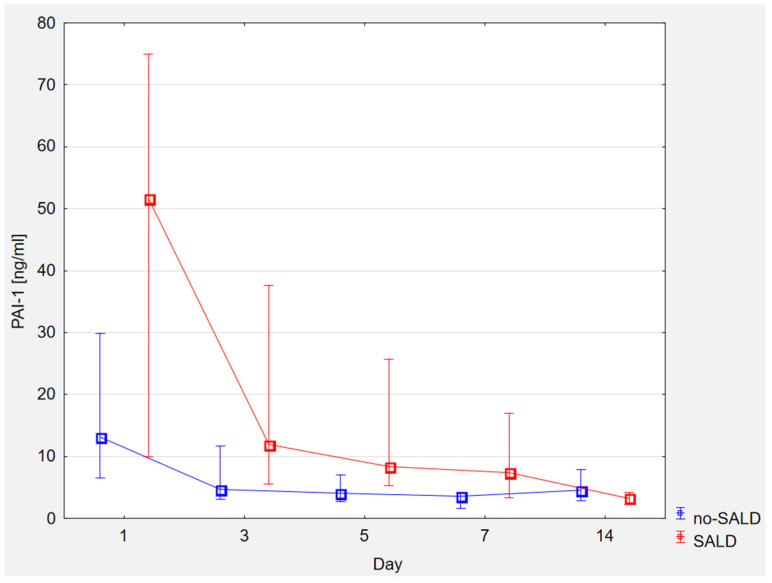
Time course of PAI-1 levels in the SALD and no-SALD groups.

**Figure 3 ijerph-20-04846-f003:**
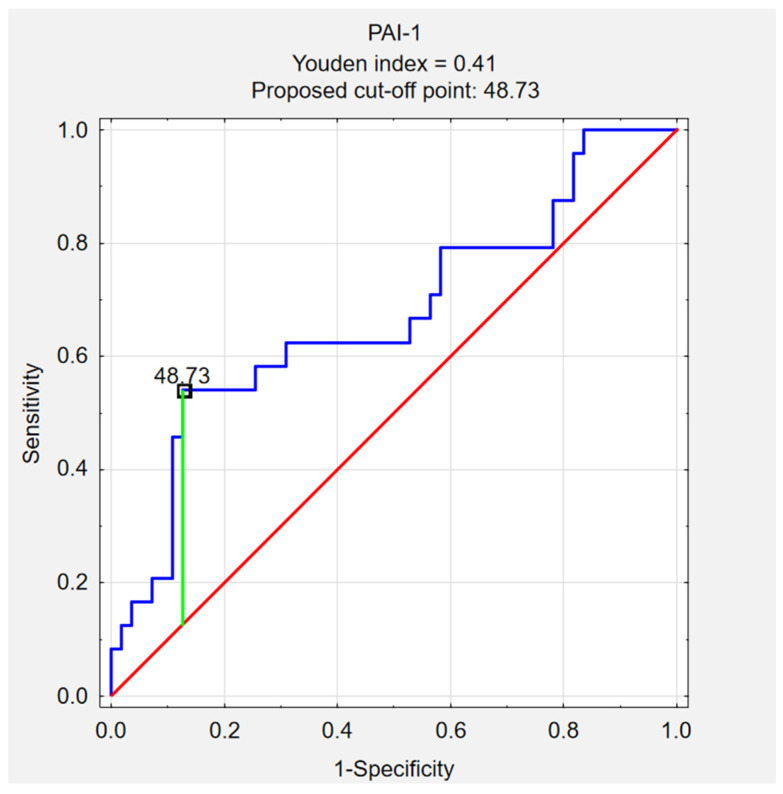
ROC curve for PAI-1.

**Figure 4 ijerph-20-04846-f004:**
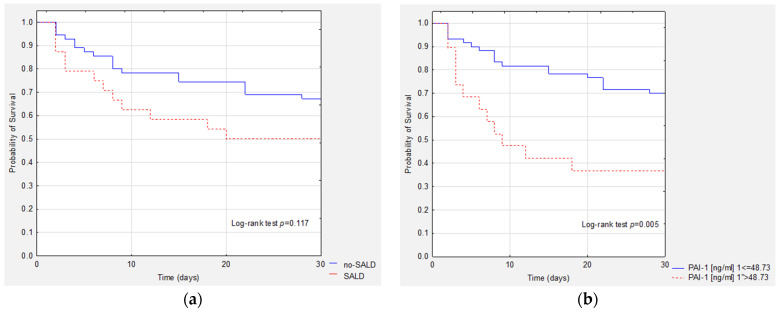
Kaplan–Meier survival curves for: (**a**) bilirubin—comparison between the SALD and no-SALD groups; (**b**) PAI-1.

**Table 1 ijerph-20-04846-t001:** Characteristics of the groups at the time of inclusion and 28 day survival.

	All Patients (n = 79)	no-SALD (n = 55)	SALD (n = 24)	*p*
Age (years)				0.63 *
Me (Q1–Q3)	66 (57–76)	67 (60–74)	61 (46–78)	
Min–Max	19–88	23–88	19–88	
Sex				0.83 **
(male/female)	31/48	22(40%)/33(60%)	9(38%)/15(62%)	
SOFA (pts.)				0.09 *
Me (Q1–Q3)	10 (7–13)	9 (7–13)	12(9–14)	
Min–Max	3–17	3–17	3–15	
APACHE II (pts.)				0.98
Me (Q1–Q3)	24 (15–30)	24 (15–30)	24 (15–28)	
Min–Max	15–30	8–44	7–37	
Sepsis/Septic shock	32/47	25 (78%)/30 (64%)	7 (22%)/17 (36%)	0.18 **
28 day survival	49 (62 ± 5.5%)	37 (67.3 ± 6.3%)	12 (50.0 ± 10.2%)	0.12 ***

Abbreviations: n, number of participants; M, mean; Me, median; Min, minimum value; Max, maximum value; Q1, lower quartile; Q3, upper quartile; p, level of statistical significance. * Mann–Whitney U test; ** Chi-square test; and *** log-rank test.

**Table 2 ijerph-20-04846-t002:** Comorbidities of the patients.

Comorbidities				
	All Patients (n = 79)	no-SALD (n = 55)	SALD (n = 24)	*p*
HTA	37 (46.84%)	26 (47.27%)	11 (45.83%)	0.89
DM	23 (29.11%)	19 (34.55%)	4 (16.67%)	0.18
CCD	3 (3.8%)	1 (1.82%)	2 (8.33%)	0.22
CA	8 (10.13%)	6 (10.91%)	2 (8.33%)	0.99
OBS	8 (10.13%)	5 (9.09%)	3 (12.5%)	0.69
CKD	18 (23.68%)	15 (27.27%)	3 (12.5%)	0.15
CPD	10 (12.66%)	8 (14.55%)	2 (8.33%)	0.72
ARY	16 (20.25%)	11 (22%)	5 (20.83%)	0.99
ALC	10 (12.66%)	7 (12.73%)	3 (12.5%)	0.99
NEUR	4 (5.06%)	4 (7.27%)	0	0.31
OTH	8 (10.13%)	5 (9.09%)	3 (12.5%)	0.69

Abbreviations: HTA: hypertension arterial; DM: diabetes mellitus; CCD: chronic cardiac disease; CA: cardiac arrest; OBS: obesity; CKD: chronic kidney disease; CPD: chronic pulmonary disease; ARY: arrythmia; ALC: alcoholic disease; NEUR: neurological disease; and OTH: other.

**Table 3 ijerph-20-04846-t003:** Source of sepsis.

Source of Sepsis				
	All Patients (n = 79)	No-SALD (n = 55)	SALD (n = 24)	*p*
Abdominal infection (ABD)	32 (40.51%)	23 (41.82%)	9 (37.5%)	0.91
Pulmonary infection (PNEU)	27 (34.18%)	17 (30.91%)	10 (41.67%)	0.50
Soft tissue infection (TISS)	3 (3.8%)	2 (3.64%)	1 (4.17%)	0.99
Neuroinfection (NEUR)	4 (5.06%)	3 (5.45%)	1 (4.17%)	0.99
Bloodstream infection (BSI)	3 (3.8%)	0	3 (12.5%)	0.026
Urinary tract infection (UTI)	6 (7.59%)	6 (10.91%)	0	0.17
UTI. PNEU	1 (1.27%)	1 (1.82%)	0	0.99
NEUR. BSI	1 (1.27%)	1 (1.82%)	0	0.99
PNEU. BSI	2 (2.53%)	2 (3.64%)	0	0.99

**Table 4 ijerph-20-04846-t004:** Routinely measured biomarkers.

	Normal Plasma Values	no-SALD (n = 55)	SALD (n = 24)	*p*
		Me (Q1–Q3)	Me (Q1–Q3)	
AST (U/L)	5–34	51 (29–155)	124 (38–325)	0.020
ALT (U/L)	0–55	46 (20–81)	61.5 (27–102)	0.16
Albumin (g/dL)	3.2–4.6	2.35 (2.0–2.65)	2.3 (1.8–2.7)	0.85
AT III (%)	80–120	58.5 (45–81.1)	53.3 (41.3–61.3)	0.08
INR	0.9–1.3	1.26 (1.12–1.45)	1.35 (1.16–1.83)	0.13
Prothrombin ratio (%)	80–114	79.7 (66.6–87.8)	73.7 (55.8–86.3)	0.27
APTT (s)	21–30.1	36.9 (30.9–46.7)	44.3 (33.9–53.6)	0.09
ALP (U/L)	<270	75 (54–148)	147.5 (94–206)	0.003
Lactate (mmol/L)	<2	2 (1.3–3.6)	4.75 (1.6–8.95)	0.041
PCT (ng/mL)	<0.05	4 (1.7–17.8)	28 (3.1–71.3)	0.0144
BIL (mg/dL)	0.3–1.2	0.70 (0.50–1.30)	2.25 (1.10–4.16)	<0.001
CRP (mg/L)	<5	239.8 (141.8–322.9)	247.2 (155–318)	0.44

Abbreviations: n, number of participants; Me, median; Q1, lower quartile; Q3, upper quartile; p, level of statistical significance; AST, aspartate aminotransferase; ALT, alanine aminotransferase; AT III, antithrombin III; INR; international normalised ratio; APTT, activated partial thromboplastin time; ALF, alkaline phosphatase; CRP, C-reactive protein; PCT, procalcitonin; and BIL, bilirubin.

**Table 5 ijerph-20-04846-t005:** Analysed biomarkers.

	Normal Plasma Values	All Patients (n = 79)	no-SALD (n = 55)	SALD (n = 24)	*p*
IP-10 (pg/mL)	47–382	932.7 ± 1643.5	321.9 (198.7–796.9)	315.4 (140.7–961.1)	0.99
ET-1 (pg/mL)	0.58–1.96	2.06 ± 1.60	1.72 (1.19–2.45)	1.54 (1.06–2.39)	0.79
PAI-1 (ng/mL)	0.99–16.9	38.3 ± 53.3	13.1 (6.5–29.9)	51.6 (9.9–74.9)	0.020
TAT (ng/mL)	0.5–10	21.3 ± 116.6	4.3 (2.6–6.5)	3.75 (2.83–5.75)	0.81
Hepcidin (ng/mL)	82.4–56700	221.8 ± 167.1	195 (101.1–321.9)	158.2 (93.5–217.9)	0.35

Abbreviations: IP-10, interferon-gamma inducible protein 10 kDa; ET-1, endothelin-1, hepcidin; PAI-1, plasminogen activator inhibitor 1; and TAT, thrombin-antithrombin.

**Table 6 ijerph-20-04846-t006:** Univariate and multivariate Cox regression analysis of risk factors influence on mortality.

Variables	Univariate			
	HR	95% CI		*p*
Age	1.04	1.01	1.07	0.011
SOFA	1.14	1.03	1.27	0.009
APACHE II	1.06	1.02	1.10	0.003
Lactate (mmol/L)	1.14	1.06	1.22	<0.001
CRP (mg/L)	1.00	1.00	1.00	0.53
PCT (ng/mL)	1.00	1.00	1.00	0.98
BIL (mg/dL)	1.12	0.94	1.35	0.21
AST (U/L)	1.00	1.00	1.00	0.82
ALP (U/L)	1.00	1.00	1.00	0.96
PAI-1 (ng/mL)	1.01	1.00	1.01	0.007
Sex (ref. M)	1.78	0.87	3.65	0.11
SALD (ref. no)	1.80	0.87	3.75	0.12
PAI-1 (ref. ≤ 48.73)	2.97	1.42	6.19	0.004
	Multivariate			
Age	1.04	1.02	1.07	1.002
Lactate (mmol/L)	1.18	1.09	1.29	<0.001

Abbreviations: HR, hazard ratio; CI, confidence interval; SOFA, the Sequential Organ Failure Assessment score; APACHE II, the Acute Physiology and Chronic Health Evaluation II score; SAPS II; BIL, bilirubin; AST, aspartate aminotransferase; CRP, C-reactive protein; PCT, procalcitonin; ALP, alkaline phosphatase; and PAI-1, Plasminogen Activator Inhibitor 1.

## Data Availability

The data presented in this study are available on request from the corresponding author.

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
