# Peer review of "The Role of Plasminogen Activator Inhibitor 1 in Predicting Sepsis-Associated Liver Dysfunction: An Observational Study"

_ijerph, 2023, doi:10.3390/ijerph20064846_

Round 1
Reviewer 1 Report
Woźnica-Niesobska et al. describes the role of PAI-1 as a potential biomarker for prognosis of Sepsis-associated liver dysfunction. This observational study is well performed and well presented. There are several other articles that have previously published the role of PAI-1 as a prognostic marker of mortality in sepsis (Hoshino, K., Kitamura, T., Nakamura, Y. et al. Usefulness of plasminogen activator inhibitor-1 as a predictive marker of mortality in sepsis. j intensive care 5, 42 (2017). https://doi.org/10.1186/s40560-017-0238-8) and the role of PAI-1 https://doi.org/10.1016/j.thromres.2005.06.007. These previous findings have not been discussed.
In Page 11, reference to Hoshino et al (32) is not reflected in the reference list. Please check your reference list.
How many samples are represented in Figure 2? Is it possible to present error bars?
Reviewer 2 Report
Dear Authors,
Many thanks for the opportunity to read your work.
The introduction explains well the reason for the study
I'm not convicted about the methodology:
You have reported the P Spearman correlation and severity of the patients (correlation demonstrated an association, but not a prognostic role), then you described a COX model: It seems "The most severe are risky patients"
I think to understand what you want to do or demonstrate: from my point of view you should postulate and report a power analysis of the study,
Then you should search a prognostic role using probit and ROC CURVE model and report all positive predicting values, predicting negative values, sensitivity or specificity
The study should be easy to understand and readable: "Less is more."
Discussion: evaluate change in case of different results
Conclusion: consider to change in case of different results
Reviewer 3 Report
In this manuscript, Woźnica-Niesobska et. al., assessed a number of plasma biomarkers in Sepsis-associated liver dysfunction (SALD) and identified Plasminogen Activator Inhibitor 1 (PAI-1) as a potential biomarker in SALD development. While the study is interesting, recent studies also concluded PAI-1 as a potential biomarker (eg. PMIDs: 29967603, 28702197, 33173586, 32166258) in Sepsis. Surprisingly, authors didn't cite many of these references.
While considering a number of biomarkers in this study, authors have excluded inflammation biomarker (C-reactive protein). Do authors have an explanation for this?
In addition to the all patients as a single group (as in Table 1), it is worth considering a separate table for male and female patients. Also, any key differences must be mentioned in the results and/or discussion.
Authors have excluded ~96 samples for reasons of other causes of hyperbilirubinemia or other causes of increased ALP. However, it is worth testing PAI-1 levels in these patient plasma. Do authors have data on it? If yes, it would be helpful if provided as supplement.
Minor:
Typo in representing P values throughout the manuscript. Authors should pay attention on it.
Round 2
Reviewer 2 Report
Dear authors, you followed my suggestions
Let me give only a last suggestion, try to a different way to report table 6
Empty spaces are not good to see. Anyway report all these data.
The paper has been implemented
